# Structure and functional implications of WYL domain-containing bacterial DNA damage response regulator PafBC

Andreas U. Müller [1], Marc Leibundgut[1], Nenad Ban [1] & Eilika Weber-Ban [1]*

In mycobacteria, transcriptional activator PafBC is responsible for upregulating the majority of genes induced by DNA damage. Understanding the mechanism of PafBC activation is impeded by a lack of structural information on this transcription factor that contains a widespread, but poorly understood WYL domain frequently encountered in bacterial transcription factors. Here, we determine the crystal structure of *Arthrobacter aurescens* PafBC. The protein consists of two modules, each harboring an N-terminal helix-turn-helix DNA-binding domain followed by a central WYL and a C-terminal extension (WCX) domain. The WYL domains exhibit Sm-folds, while the WCX domains adopt ferredoxin-like folds, both characteristic for RNA-binding proteins. Our results suggest a mechanism of regulation in which WYL domain-containing transcription factors may be activated by binding RNA or other nucleic acid molecules. Using an in vivo mutational screen in *Mycobacterium smegmatis*, we identify potential co-activator binding sites on PafBC.

[1] ETH Zurich, Institute of Molecular Biology and Biophysics, CH-8093 Zurich, Switzerland. *email: eilika@mol.biol.ethz.ch

DNA damage represents a threat to the integrity of genetic information and is therefore counteracted in all organisms by an arsenal of DNA repair processes that are activated by specific DNA damage response pathways. Mycobacteria and many other actinobacteria employ two distinct yet interconnected pathways in order to upregulate the expression of specific sets of genes required for repair and survival of DNA damage.

The "SOS response", the canonical pathway described in most bacterial species, relies on cleavage and removal of LexA, a transcriptional repressor of DNA repair genes ("SOS genes") (reviewed in refs. [1,2]). Under normal conditions, LexA ensures low expression levels of the SOS genes by binding to a promoter motif called "SOS box"[3]. Single-stranded DNA (ssDNA) fragments accumulating under DNA damage conditions serve as DNA stress signal for the SOS response and are sensed by the ATPase RecA. RecA forms a filamentous complex with ssDNA that is able to induce autoproteolytic cleavage of the LexA repressor, leading to derepression of the SOS genes[4–6]. In *Mycobacterium tuberculosis* (Mtb), the LexA repressor controls about 25 genes[7,8].

In contrast, the second pathway regulates over 150 genes, including many of the LexA-controlled genes, amongst them also *recA*, the DNA damage sensor and co-regulator of the SOS response. This other pathway operates independently of LexA and RecA, as demonstrated by deletion of *recA* in Mtb, which leaves upregulation of most DNA repair genes intact[9,10]. Different from the regulatory principle of derepression, these genes are regulated by transcriptional activation by the heterodimeric protein complex PafBC[11,12]. The complex consists of the close sequence homologs PafB and PafC (proteasome accessory factors B and C) that are encoded together in an operon that is tightly associated with the bacterial proteasome gene locus, suggesting a functional connection. Indeed, many DNA repair proteins are removed by proteasomal degradation after the DNA damage has been repaired, thereby helping to shut down the stress response and preventing negative impact of DNA-modifying activities under normal conditions[12].

PafBC activates its target genes via a promoter motif called RecA-NDp (RecA-independent promoter), which was demonstrated by in vivo identification of PafBC binding sites using cell culture cross-linking followed by immunoprecipitation of PafBC-DNA complexes[12]. However, PafBC protein levels are not changing in response to DNA stress[11]. Furthermore, specific interaction between PafBC and the identified DNA target regions could not be reconstituted in vitro. Taken together, these results suggest that an additional "response-producing" event must take place to initiate PafBC transcription activation.

In order to establish the mechanistic principles employed by PafBC to activate transcription at the molecular level, understanding of the structural framework is crucial. Based on sequence similarity, PafBC belongs to a family of bacterial regulators featuring a winged helix-turn-helix (HTH) domain at the N-terminus, followed by a C-terminal domain of unknown function named WYL domain after a consecutive W-Y-L sequence motif. It has been suggested that the WYL domain might play the role of a ligand-binding domain in the context of this class of transcription factors. A handful of other WYL domain-containing proteins were studied to date: (1) DriD, an SOS response-independent transcriptional activator of a cell division inhibitor protein in *Caulobacter crescentus*[13] (2) Sll7009, Sll7062, Sll7078, transcriptional repressors of CRISPR/Cas system mature crRNA in *Synechocystis* 6803[14], (3) PIF1 helicase from *Thermotoga elfii*[15] and (4) WYL domain-containing proteins stimulating RNA cleavage by Cas13d in *Eubacterium siraeum* and *Ruminococcus sp*[16]. However, structural information on WYL domain-containing transcriptional regulators is missing, and evidence as to how they exert their functions mechanistically has remained elusive.

In this study, we determine the crystal structure of PafBC from *Arthrobacter aurescens* in its non-activated, DNA-free state. The structure reveals that the WYL domain exhibits an Sm-fold, commonly encountered in RNA-binding proteins, and is followed by an additional C-terminal extension (WCX) domain featuring a ferredoxin-like fold. Based on the structure of the PafBC WYL-domain, we carry out a comprehensive computational analysis of WYL domain-containing proteins, and demonstrate that the WYL domain is a widespread feature of bacterial transcription factors present in almost all bacterial taxa. Our study shows that Sm-fold proteins are a much more frequent occurrence in bacteria than previously thought. Based on the high structural similarity of the WYL motif-containing domain to the bacterial RNA chaperone Hfq and the known binding sites of Hfq, we identify functionally essential residues in the WYL domain of PafBC, which are likely involved in binding of a response-producing ligand in this distinct class of transcriptional regulators.

## Results

**The crystal structure of PafBC reveals an asymmetric conformation.** In order to obtain information about the architecture of the PafBC class of transcriptional regulators, we set out to determine the crystal structure of PafBC. We carried out crystallization experiments using a range of PafBC orthologs from different actinobacterial organisms, also including PafBC proteins from organisms encoding a naturally fused PafBC complex (i.e., from *Kocuria rhizophila*, *Thermobifida fusca*, and *Arthrobacter aurescens*). Three-dimensional crystals suitable for data collection were ultimately obtained using an *A. aurescens* PafBC construct (AauPafBCΔNC), which was shortened by 17 amino acids at the N-terminus and 7 amino acids at the C-terminus based on sequence alignment, since these residues are not conserved amongst the orthologs and not even present in most of them. Indeed, it is likely that the start site for the *A. aurescens* protein was misassigned, since a valine (encoded by GTG) is present at the position where the other PafBC proteins feature the conserved initiator methionine, and GTG is a frequent start codon in actinobacteria[17,18] (Supplementary Fig. 1). Selenomethionine-labeled protein was used for crystallization and the structure was determined de novo by single-wavelength anomalous diffraction. AauPafBCΔNC crystallized in space group $P2_12_12_1$ with two molecules in the asymmetric unit, and the structure was refined to 2.2 Å with an $R_{work}/R_{free}$ of 20%/24% (Table 1). The structural model was built to near completion, encompassing 1279 residues out of 1328. The missing residues are all located in three poorly ordered loop regions. Although AauPafBC is a natural fusion protein and thus consists of a single polypeptide chain, for simplicity and easier comparison to the majority of actinobacteria encoding separate PafB/PafC proteins we will refer to the PafB-corresponding and PafC-corresponding parts as PafB or PafC, respectively.

Although most transcriptional regulators with an HTH domain feature an internal symmetry axis when bound to DNA[19], PafBC in our structure adopts an asymmetric conformation (Fig. 1, Supplementary Fig. 2 and Supplementary Fig. 3). The asymmetric arrangement of PafBC is not surprising, since PafBC is not in complex with its consensus DNA binding site and its domain arrangement reflects the non-activated form of PafBC. This is in agreement with the observation that specific interaction between PafBC and the identified DNA target regions takes place in vivo, but could not be reconstituted in vitro[11,12], further supporting the

**Table 1 Data collection and refinement statistics of the crystal structure of *Arthrobacter aurescens* PafBC (^AauPafBCΔNC)**

| | ^AauPafBCΔNC (SeMet), PDB ID 6SJ9 |
|---|---|
| **Data collection** | |
| Wavelength | 0.978561 Å (12,670 eV) |
| Resolution range | 48.72−2.2 (2.279−2.2) |
| Space group | P2₁2₁2₁ |
| Unit cell dimensions (Å) | 77.88, 119.03, 160.21 |
| α, β, γ (°) | 90, 90, 90 |
| Total reflections | 5,208,738 (520,414) |
| Unique reflections | 76,285 (7554) |
| Multiplicity | 68.3 (68.9) |
| Completeness (%) | 99.95 (99.96) |
| Mean I/sigma(I) | 31.58 (2.76) |
| Wilson B-factor | 48.04 |
| R-merge | 0.1226 (2.069) |
| R-meas | 0.1235 (2.084) |
| R-pim | 0.01484 (0.2494) |
| CC1/2 | 1 (0.871) |
| CC* | 1 (0.965) |
| Matthews coefficient | 2.49 |
| Solvent fraction | 0.505 |
| Molecules/ASU | 2 |
| **Refinement** | |
| Reflections used in refinement | 76,278 (7555) |
| Reflections used for R-free | 3814 (378) |
| R-work | 0.2032 (0.2884) |
| R-free | 0.2374 (0.3351) |
| CC(work) | 0.957 (0.833) |
| CC(free) | 0.938 (0.793) |
| Number of TLS groups | 10 |
| **Model statistics** | |
| Number of non-hydrogen atoms | 10,249 |
| -in macromolecules | 9936 |
| -in ligands | 49 |
| -in solvent | 264 |
| Protein residues | 1279 |
| Heavy atom sites (Se) | 14 |
| RMS(bonds) | 0.012 |
| RMS(angles) | 1.23 |
| Ramachandran favored (%) | 98.10 |
| Ramachandran allowed (%) | 1.90 |
| Ramachandran outliers (%) | 0.00 |
| Rotamer outliers (%) | 1.45 |
| Clashscore | 5.94 |
| Average B-factor | 74.60 |
| -macromolecules | 75.11 |
| -ligands | 88.63 |
| -solvent | 52.48 |

notion that PafBC is in a non-activated state in absence of the putative DNA stress-sensing ligand.

The PafBC structure features six distinct domains, three in each of the homologous PafB and PafC modules. Each module includes an N-terminal HTH domain followed by two other domains (Fig. 1 and Supplementary Fig. 2). To distinguish between the homologous domains of each module, we refer to the domains belonging to the PafB module as helix-turn-helix (HTH-B), WYL (WYL-B), and C-terminal extension of the WYL (WCX-B) domains, while in PafC the corresponding domains are termed HTH-C, WYL-C, and WCX-C, respectively. All individual domains are connected by long loops, which are particularly pronounced between the WYL and WCX domains, suggesting a high degree of flexibility and conformational adaptability of the PafBC complex.

Both HTH domains adopt a classical winged helix-turn-helix fold with a three-helix bundle consisting of helices H1, H2 and H3 followed by a two-stranded wing, a topology that is typical for many transcriptional regulators (Fig. 2 and Supplementary Fig. 4a). From structures of other winged HTH domains in complex with DNA, it is known that H3 usually mediates the specific DNA recognition by binding into the major groove, while the wing provides additional contacts in the minor groove[20–22]. The putative recognition helix of PafB features two highly conserved phenylalanines (F42 and F46) involved in forming the hydrophobic core of the three-helix bundle (Supplementary Fig. 4b). From the opposing surface-accessible side of the recognition helix, two strictly conserved arginine side chains project outwards (R44 and R48). These residues might be involved in sequence-specific DNA binding with guanine and cytosine bases. Notably, the putative recognition helix of PafC is shorter and comprises a different amino acid sequence that would suggest recognition of non-palindromic DNA sequence, which is in accordance with the non-palindromic nature of the PafBC binding motif (RecA-NDp)[12,23]. However, the most striking difference between the HTH domains in PafB and PafC is their location and accessibility in the solved complex structure.

The HTH domain of PafB (HTH-B) is accessible in the structure, since only helix H1 and a small part of H3 make hydrophobic contacts to the rest of the molecule (Supplementary Fig. 4b). In contrast, the HTH domain of PafC (HTH-C) interacts extensively with the other domains of the protein (Supplementary Fig. 2b and Supplementary Fig. 4c–f). Importantly, the putative recognition helix (H3) appears wedged into the protein core, and following H3, a long loop extends around the back of the central α-helix of PafB (α4) and the WYL-B domain (Fig. 2b and Supplementary Fig. 4c). Superposition of the HTH-B and HTH-C domains reveals a good agreement of the folds except for a much longer β1/β2-loop (wing) in HTH-B, while HTH-C displays a very short connection between β1 and β2 and instead features a long loop between H3 and β1 (Supplementary Fig. 4a). Sequence alignment-assisted comparison of the structural elements reveals that the β2 strands of both HTH domains occupy the same position relative to the helices, while the β1 position differs. This opens up the possibility that the wing of HTH-C has undergone a register shift to accommodate HTH-C in the protein core of the non-activated PafBC complex (Supplementary Fig. 5).

In all PafB and PafC homologs, the HTH domains are connected to the WYL domains via a sequence stretch of roughly 30 residues. This region forms a long single helix in the PafB module of our structure. The helix is located in the core of the PafBC structure, traversing the entire complex (Fig. 1 and Supplementary Fig. 3). In contrast, the equivalent region in PafC consists of three smaller helices forming a bundle between HTH-C and WYL-C (α4', α4", α4'", Supplementary Fig. 2). The long central helix of PafB packs against helix H1 of the HTH domain in PafC (Supplementary Fig. 4c). The WYL domains of PafB and PafC are located at the perimeter on opposite sides of the complex.

**The PafBC WYL domains feature an Sm-fold**. The structure shows that the domain previously referred to as the "WYL domain" of PafB or PafC in fact consists of two separate domains, with only the first domain featuring the characteristic WYL sequence motif. In the context of this study we refer to this domain as the WYL domain and to the second domain as WCX domain. The PafBC WYL domains are located at the periphery of the molecule on opposite sides, while the WCX domains come together to form a dimeric interaction module.

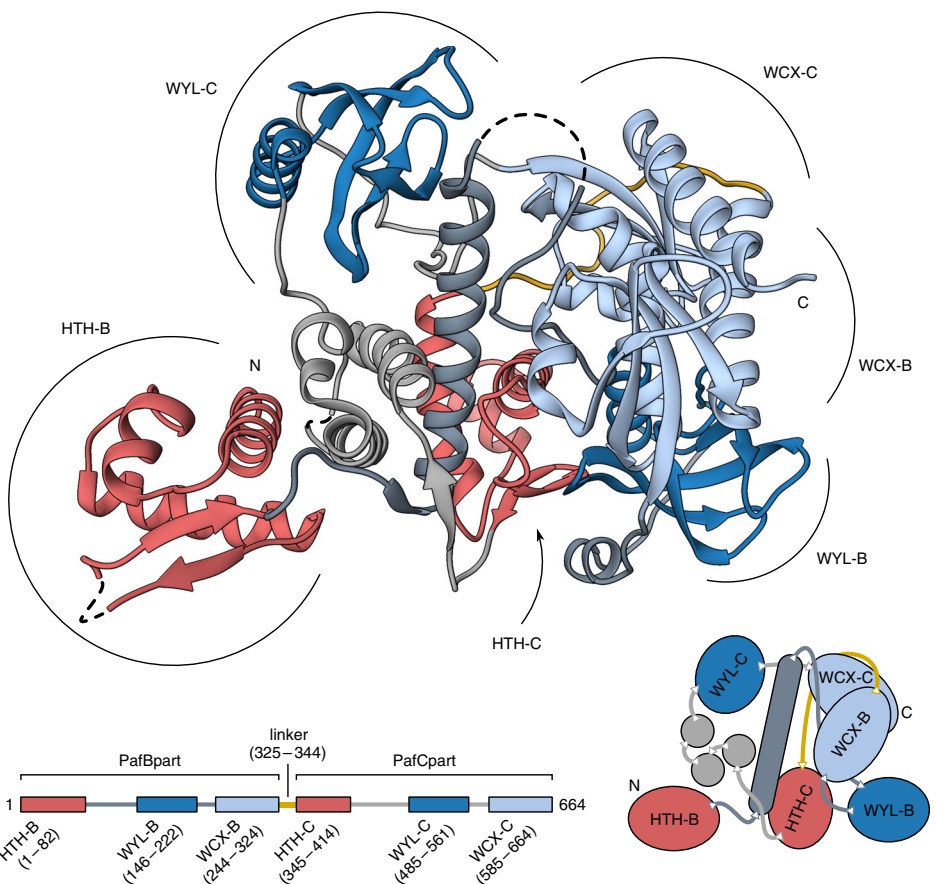

**Fig. 1** Crystal structure of the PafBC complex from *Arthrobacter aurescens* ($^{Aau}$PafBCΔNC). The protein is the product of a natural gene fusion and encompasses a PafB and a PafC part (bottom scheme) connected via a 20 amino acids long linker (yellow). The winged helix-turn-helix domains (HTH) are colored in red, WYL domains are colored in blue and the C-terminal extension of the WYL domain (WCX) is colored in light blue. Other parts of the structure belonging to PafB are colored in dark gray, while the remaining PafC parts are colored in light gray. Dashed lines bridge gaps in the model. See also Supplementary Fig. 1, Supplementary Fig. 2, and Supplementary Fig. 3

Notably, in this interaction module the WXC domains are arranged in a two-fold pseudo-symmetric manner, in spite of the overall asymmetric arrangement of the domains in the PafBC structure (Fig. 3a, b). Each WCX domain harbors a four-stranded anti-parallel β-sheet of 4-1-3-2 topology framed by two short α-helices, which contain a hydrophobic core (Fig. 3b, c left and Supplementary Fig. 6a). The main chain sharply bends at a highly conserved cis-proline into a C-terminal α-helix, which crosses the C-terminal α-helix of the other WCX domain (Supplementary Fig. 6b). Interaction of the WCX domains arises through interdigitation of two pairs of helices. The contacts are stabilized by salt-bridges, hydrogen bonds and a conserved hydrophobic island containing a pair of highly conserved leucines (Supplementary Fig. 6b–e). In fact, the majority of hydrogen bonding occurs at the ends of the crossed C-terminal α-helices, while high conservation among the interacting residues seems to be restricted to the two leucines (Supplementary Fig. 6b). The PafBC interaction via the WCX domains represents a strong element in the PafBC non-covalent interaction and is probably maintained also in the active DNA-binding form.

We carried out a comparative analysis based on the protein fold of the WYL and WCX domains using the Dali fold recognition program to discover structural homologs[24]. The search using the isolated WCX domain yielded a very broad spectrum of hits ranging from spliceosomal proteins, elongation factors, oxidoreductases to proteases. After further manual assessment of the individual hits, we discovered that the WCX

domain follows a typical ferredoxin-like fold (βαββαβ) with an ancillary C-terminal α-helix. Interestingly, a number of RNA-binding proteins such as human hnRNP A1 (Fig. 3c, middle), CRISPR/Cas protein Cse3 (Fig. 3c, right) and ribosomal proteins also contain the ferredoxin-like fold and these domains are directly involved in RNA interaction (Supplementary Fig. 7).

The fold of the WYL domain consists of a five-stranded anti-parallel β-sheet with a 5-1-2-3-4 topology preceded by an α-helix (Fig. 4a). The strands are strongly curved and the middle β2-strand is almost twice the length of the other five, causing it to arch back over itself and resulting in a β-sandwich topology, where β-strands 5-1-2 make up one half and strands 2-3-4 the other. Middle strand β2 is participating in both and connects the two halves. The eponymous WYL residues are located in β3, with the highly conserved tyrosine pointing away from the hydrophobic core. Structure similarity searches using the isolated WYL domain on the Dali webserver[24] returned PDB entries of proteins containing an Sm-fold, like the bacterial RNA chaperone Hfq (host factor for RNA bacteriophage Qβ replication) and certain spliceosomal proteins. Proteins containing an Sm-fold are very abundant in eukaryotes, while only few examples (amongst them Hfq) have been described in bacteria. Closer comparison of the WYL domain with Hfq shows that the WYL domain Sm-fold features a slightly longer N-terminal helix, longer β2, and β3 strands, as well as longer loops between strands β1/β2 and β3/β4 (Fig. 4a, b). Many proteins of the Sm-like family were shown or predicted to bind RNA. The structural similarity of the

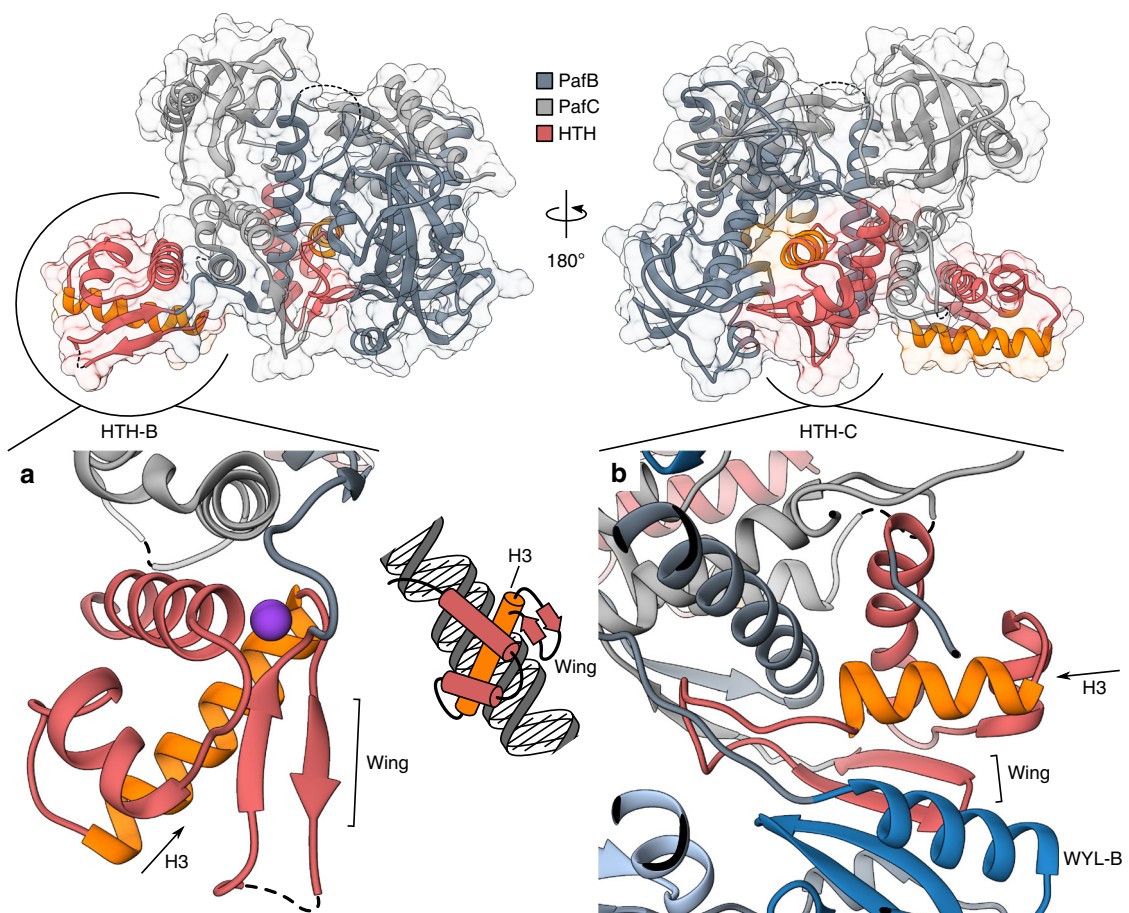

**Fig. 2** The non-activated conformation of PafBC buries the HTH domain of PafC. **a** The HTH domain of PafB (HTH-B, red) adopts a typical winged helix-turn-helix fold (wHTH). The recognition helices (H3) of wHTH domains typically insert into the major groove of the DNA, while the wings establish contact to the minor groove (small inset, middle). The recognition helix of HTH-B (orange) is exposed and available to accept a DNA ligand. The main chain of H1 and the wing coordinate a potassium ion (lilac sphere). **b** The HTH domain of PafC (HTH-C, red) on the other hand forms part of the protein core. H3 of HTH-C (orange) is shorter by two helical turns compared to H3 of HTH-B, and an unstructured loop reaches into the protein core. In addition, the β-sheet of the wing extends the β-sheet of the WYL domain of PafB (WYL-B, blue). Dashed lines bridge gaps in the models. See also Supplementary Fig. 4

WYL domain to Hfq and other Sm-like proteins therefore suggests that the WYL domains provide a binding site for an RNA molecule.

**The WYL domain Sm-2 loop contains essential residues for PafBC function.** Previously, we showed that PafBC levels do not change under stress conditions and we could not detect any specific DNA binding activity towards the RecA-NDp motif in vitro[11,12], suggesting that PafBC requires a co-activator for its activity, which is only present during stress conditions. The structural homology between the WYL domains of PafBC and Sm-folds involved in RNA binding indicates that the response-producing ligand might be an RNA molecule and the WYL domain could act as a ligand-sensing domain.

In order to deduce potential ligands and ligand-binding locations from the homology between the PafBC WYL domains and the Sm-fold-containing bacterial RNA chaperone Hfq, we carried out a comparative analysis of potential binding regions. Hfq forms a homohexameric ring-shaped complex that was shown to bind RNA at three distinct sites (Fig. 4b)[25]: the proximal site exposing the α-helices and binding sRNA and mRNA (shown with ligand in Fig. 4b); the distal site binding A-rich oligonucleotides; and the rim (also called lateral) site literally represented by the rim of the Hfq ring. There is also increasing evidence that the C-terminus is functionally involved in RNA

binding[26,27]. Given the structural similarity of PafBC's WYL domains to Hfq, we compared sequence conservation patterns in both proteins. Hfq exhibits a highly conserved loop in its Sm-2 motif, containing residues that contact the RNA backbone at the proximal binding site (Fig. 4b–d). Such a highly conserved loop is also found at the corresponding locations in the PafBC WYL domains, where two arginine side chains point into the direction where the RNA ligand is positioned in Hfq (Fig. 4d, e). A potential ligand may be bound at this location and transduce the signal of DNA damage to PafBC, which in turn becomes activated to carry out its role as transcriptional activator.

To test the hypothesis that the conserved sequence stretch between strands β4 and β5 in the PafBC WYL domains has functional significance, we complemented the *M. smegmatis* Δ*pafBC* strain with PafBC mutants featuring amino acid substitutions at this location and assessed the viability of the mutants in presence of the DNA-damaging agent mitomycin C (MMC) (Fig. 5). We also chose residues at other sites in the WYL domain based on sequence conservation and structural similarity to Hfq. Specifically, we selected the conserved tyrosine that is part of the WYL triplet, another conserved tyrosine (sometimes histidine) in strand β1, and the conserved patch between β4 and β5 for mutation. The chosen residues were mutated to alanine and the mutations were introduced separately into PafB or PafC or into both proteins simultaneously. To reduce the permutation space, we decided to treat the two arginine residues in the β4/β5

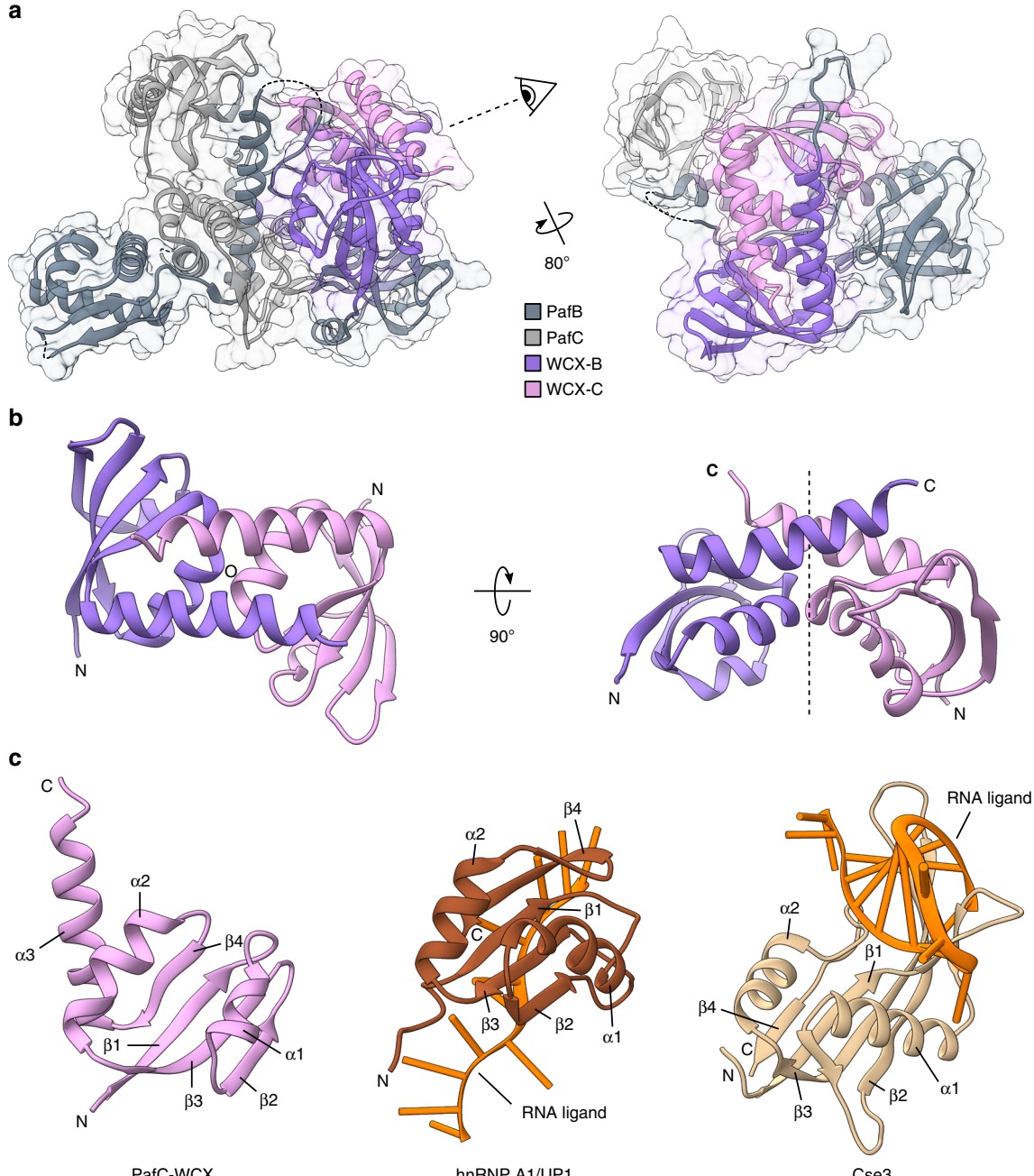

**Fig. 3** The C-terminal extension (WCX) domains of PafB and PafC contain a ferredoxin-like fold. **a** The WCX domains of PafB (WCX-B, lilac) and PafC (WCX-C, pink) contact each other in the crystal structure. Dashed lines bridge gaps in the model. **b** WCX-B and WCX-C exhibit a two-fold rotational symmetry axis (circle and dashed line). **c** The WCX domains (left; shown for PafC; residues 585–664) contain a ferredoxin-like fold with an additional C-terminal α-helix (α3). Very versatile and present in proteins with highly diverse functions, the ferredoxin-like fold is also found in many RNA-binding proteins such as human hnRNP A1 (also known as UP1; middle; brown; PDB 6DCL; residues 7–89 shown) or the C-terminal domain of CRISPR-Cas protein Cse3 (right; beige; PDB 2Y8W; residues 90–211 shown). RNA ligands are colored in orange

loop as functionally redundant (i.e., they were simultaneously substituted with alanine). Since the HTH domain of PafC would not be able to bind DNA in the observed conformation (Fig. 1 and Fig. 2b), we also deleted the HTH domains individually to establish if they are required for PafBC function.

To assess the viability of the PafBC mutant strains, the cells were first grown for a defined period of time in presence of increasing concentrations of MMC. Subsequently, the dye resazurin was added, which is reduced by living (but not by dead) cells to resorufin, giving rise to a color change. Wild type *M. smegmatis* cells grow in presence of up to 100 ng/ml MMC,

while the Δ*pafBC* strain shows growth only up to about 8 ng/ml MMC, which is in agreement with the previously determined minimal inhibitory concentrations for these strains (Fig. 5a)[11].

Complementation of the *pafBC* knockout strain with wild type PafBC restores the viability to the level observed for the wild type. Deletion of either the HTH domain of PafB or PafC leads to the same reduced viability as observed for the Δ*pafBC* strain (Fig. 5a). However, it has to be noted that the expression of ΔHTH-B was barely detectable and may therefore not be sufficient for complementation (Supplementary Fig. 8). Hence, we addressed the requirement for HTH-B by mutating two arginines in the

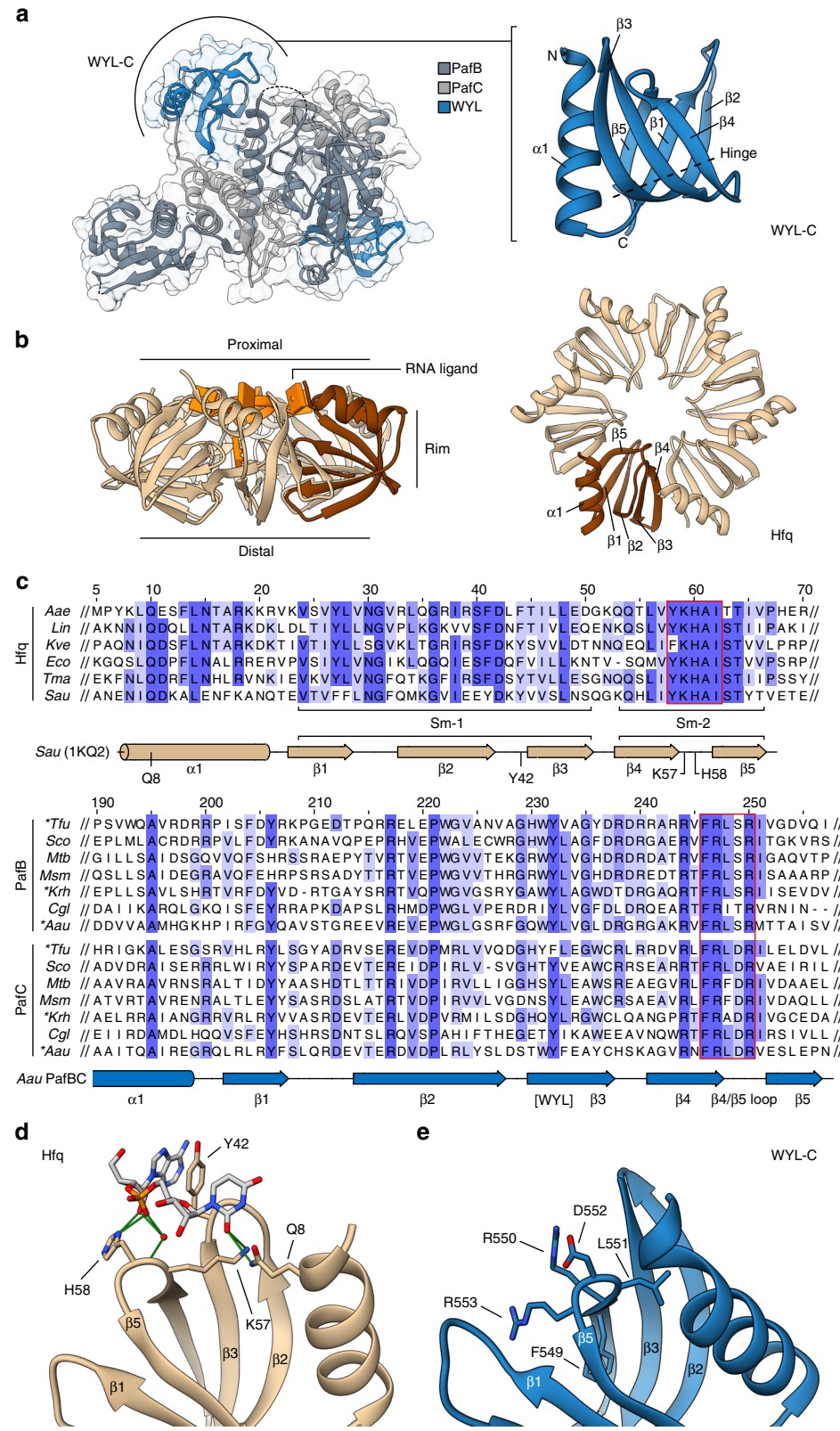

recognition helix of HTH-B (R46 and R50; equivalent to R44 and R48 in <sup>Aau</sup>PafBC, see above) to alanines, which resulted in the same phenotype as for the ΔpafBC strain (Supplementary Fig. 9a). The ΔHTH-C mutant expresses well, and the complementation experiment demonstrates that the second HTH domain (HTH-C), which in our structure is in an inaccessible conformation for DNA binding, is required for a fully functional PafBC complex.

We then tested the alanine point mutants for complementation. All 15 mutants, with the exception of double mutant B-F210A/C-F200A, exhibited similar PafBC expression levels to the wild type strain (Supplementary Fig. 8 and Supplementary Fig. 9b). The level of B-F210A/C-F200A was barely detectable, which could be either because the level is too low for detection and thus may not be sufficient for complementation or because

**Fig. 4** A conserved loop responsible for RNA binding in the Sm-fold protein Hfq is also conserved in the WYL domain. **a** The structure of the WYL domain (shown for PafC; residues 485–561) is highly similar to the Sm-fold of the bacterial hexameric RNA chaperone Hfq. **b** Each of the Hfq subunits (beige; PDB 1KQ2; one subunit colored in brown) adopts the Sm-fold. The Hfq ring can bind RNA at three distinct sites (proximal, distal, and rim); here shown with an RNA ligand bound at the proximal site. The RNA ligand of Hfq is colored in orange and is not shown for the top view to visualize the fold. **c** Multiple sequence alignments of Hfq (top) or PafB and PafC protein sequences (bottom) highlight a patch of strongly conserved residues (red boxes) located in the β4/β5-loop of the Sm-fold. Secondary structure elements of *Staphylococcus aureus* (Sau) Hfq (beige; PDB 1KQ2; UniProt Q2FYZ1) and *Arthrobacter aurescens* (Aau) PafBC (blue) are shown below each alignment. Naturally fused PafBC proteins were separated into PafB and PafC parts before alignment (asterisks). Alignment is colored according to percent identity. Numbers above the sequences refer to the alignment position numbers. Aae *Aquifex aeolicus* (O66512), Lin *Leptospira interrogans* (Q8F5Z7), Kve *Koribacter versatilis* (Q1IIF9), Eco *Escherichia coli* (P0A6X3), Tma *Thermotoga maritima* (Q9WYZ6), Tfu *Thermobifida fusca* (Q47P13), Sco *Streptomyces coelicolor* (Q9RJ64, Q9RJ65), Mtb *Mycobacterium tuberculosis* (P9WIM1, P9WIL9), Msm *Mycobacterium smegmatis* (I7G3U5, A0QZ41), Krh *Kocuria rhizophila* (B2GIN6), Cgl *Corynebacterium glutamicum* (Q8NQE2, Q8NQE3). UniProt sequence identifiers in brackets. **d** Two of the highly conserved residues of the β4/β5-loop are involved in substrate binding at the proximal face of Hfq (PDB 1KQ2, one subunit shown). Hydrogen bonds are colored in green. **e** The β4/β5-loop residues of ^AauPafBCΔNC present an interface for potential ligand binding in a similar manner as Hfq

the epitope has been compromised by the mutation. For most alanine-substitution mutants a pattern could be observed (Fig. 5b–f): If the mutation is present in only one of the WYL domains, the viability is only moderately affected, but in case both WYL domains carry the mutation, the effect seems to be additive and the viability of the cells is lowered to the level of the knockout strain. This is the case for the double arginine mutants (Fig. 5c), the tyrosine of the WYL triplet (Fig. 5d), and the phenylalanine of the β4/β5 loop (Fig. 5f), the latter with the caveat that the PafBC expression level of the B-F210A/C-F200A mutant could not be determined reliably (see above). Mutation of the β1 histidine/ tyrosine leads to a comparable result, except that the decrease in viability is milder if one of the WYL domains carries the mutations, and the effect is less severe than observed for the knockout strain if both WYL domains are mutated (Fig. 5b). Furthermore, mutation of the serine/aspartate in the β4/β5 loop did not affect viability (Fig. 5e). Our observations from the resazurin viability assay were confirmed by spotting dilution series of PafBC mutant cultures onto agar plates after exposure to MMC (Supplementary Fig. 10).

To determine whether the induction of DNA repair genes is affected in these mutants, we exposed the cultures to MMC and compared RecA levels by immunoblot analysis (Fig. 5g, h and Supplementary Fig. 9c). PafBC mutants with a decrease in viability as observed in the resazurin assay also exhibited a corresponding decrease in RecA induction level, linking the phenotype to the inability to induce a proper DNA damage response.

Our results demonstrate that the conserved residues in the β4/β5 loop and β1 of the WYL domain are required for the function of PafBC, likely because they interact with a signal-transducing ligand. Furthermore, successive inactivation of the subunits has an additive effect, suggesting that there are two ligand binding sites present, one at each WYL domain. In summary, our observations show that both subunits of the PafBC complex are functional and both WYL domains are required for full viability.

**The WYL domain is mainly associated with DNA-binding domains.** Based on the structural analysis of the WYL domains in the PafBC complex and the fact that this domain occurs also in other bacterial regulators, we carried out a thorough bioinformatics analysis of WYL domain-containing proteins to understand, in which functional context they occur and how widely distributed they are.

We computationally analyzed the co-occurrence of the WYL domain with other domains along with its taxonomic distribution using hidden Markov models (HMMs)[28]. HMMs are widely used for finding distant protein homologs and they provide the basis for one of the largest protein family databases, Pfam, which

groups proteins containing the same domain into families. Our structural analysis has shown that the PafBC C-terminal part originally assigned as "WYL" domain as a whole, in fact consists of two domains, the actual WYL domain and a C-terminal extension (WCX) domain. Based on the domain boundaries of the WYL domains in our structure and sequence alignment with other PafBC orthologs, we generated a WYL domain HMM and used it to retrieve all WYL domain-containing proteins from the UniProt reference proteomes yielding 15,079 entries (Supplementary Data 1 and Supplementary Data 2). The resulting entries were distributed across 5330 different species with only 81 sequences from 50 species among *Eukaryota*, *Archaea* or Viruses, which were mostly candidate species (Supplementary Data 2). Thus, the WYL domain appears to be limited to bacteria and we restricted our subsequent analyses to bacterial sequences (Supplementary Data 1).

In order to identify domain families associated with WYL domain proteins, the retrieved WYL domain-containing bacterial sequences were annotated based on all Pfam HMMs and additional HMM profiles we generated for the WCX domain and the PafBC N-terminal winged HTH domain that was not recognized by any of the existing Pfam HMMs. Two observations can immediately be made from the final set of domain architecture classes (Fig. 6a and Supplementary Fig. 11b): First, the majority of classes, covering more than 90% of sequences, shows co-occurrence of the WYL domain with an HTH domain preceding it. Second, the WYL domain is primarily present together with a C-terminally located WCX domain, and only about 25% of sequences exhibit the WYL domain alone.

About two thirds of all sequences are found in class A, which also comprises all PafB and PafC sequences. The second largest group, class B (15% of all hits), contains proteins with only an HTH and a WYL domain, lacking the WCX domain. Notably, class C could also be viewed as a subgroup of class A, as it is made up of natural fusion proteins of actinobacterial PafB and PafC homologs. A significant number of sequences contain a Helicase C3 domain in combination with the WYL domain, but lacking the WCX domain.

The distribution of WYL domain-containing proteins among bacterial phyla reflects the distribution of these phyla in the reference proteomes, showing that WYL domains are ubiquitous among bacteria (Supplementary Fig. 11a). Interestingly, the gram-positive phyla of *Actinobacteria* and *Firmicutes* exhibit on average roughly 5.2 and 2.8 WYL domain-containing proteins per organism, respectively, while the gram-negative phyla of *Proteobacteria* and *Bacteroidetes* show only 2.0 and 2.1 average WYL domain-containing proteins per organism, respectively (Fig. 6b). By analyzing the taxonomic distribution for each domain architecture class, we observed that the PafBC-like class A is most prominently found in *Actinobacteria*, *Firmicutes*, and *Bacteroidetes*, but much

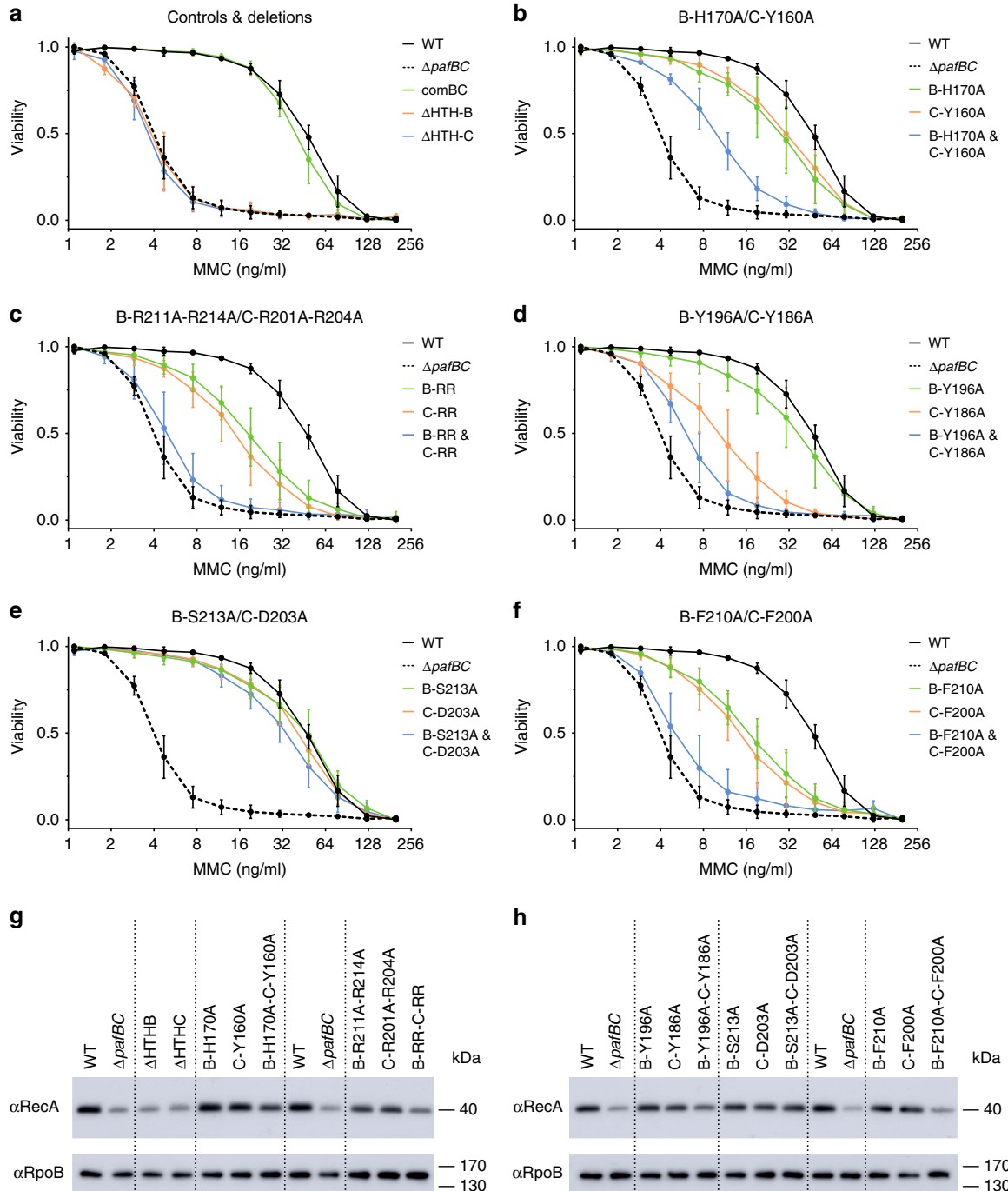

**Fig. 5** Mutational complementation screen of residues in the WYL domain of PafBC in *Mycobacterium smegmatis*. **a** The *M. smegmatis* Δ*pafBC* strain (Δ*pafBC*, dashed black line) exhibits a lower viability by an order of magnitude in presence of the DNA-damaging agent mitomycin C (MMC) than the wild type strain (WT, solid black line), which is restored by expressing wild type PafBC (comBC, green). Expression of PafBC with either the HTH domain of PafB or PafC deleted (ΔHTH-B, orange or ΔHTH-C, blue) results in the same phenotype as observed for the knockout strain. **b**–**f** Complementation with alanine substitution mutants in either PafB (green) or PafC (orange) moderately affect viability, while mutations in both proteins (blue) additively reduce the viability, in three cases to knockout levels (**c**, **d**, **f**). Each data point represents the mean of three or more individual experiments. Error bars represent the standard deviation of the mean. **g**–**h** RecA induction in the complemented strains in response to MMC exposure was compared to the knockout (Δ*pafBC*) and wild type (WT) strains carrying the empty plasmid. Strains were grown to OD600 of 1.0 and exposed to 80 ng/ml MMC for 4 h before immunoblotting for RecA. RpoB served as loading control. See also Supplementary Fig. 8 and Supplementary Fig. 9

less abundant in *Proteobacteria* (Fig. 6c). On the other hand, more than two thirds of the HTH-WYL architecture members (class B) are found in proteobacterial species (Fig. 6d).

Taken together, our analysis shows that the majority of WYL domain-containing proteins are transcriptional regulators based on the presence of an HTH domain. It therefore seems likely that the mechanism of transcriptional regulation and signal relay employed by PafBC, although currently unknown, is a widespread principle found in almost all bacteria. Moreover, in some phyla multiple of these transcriptional regulators are present in one organism, suggesting that WYL domain-containing regulators may be involved in different pathways.

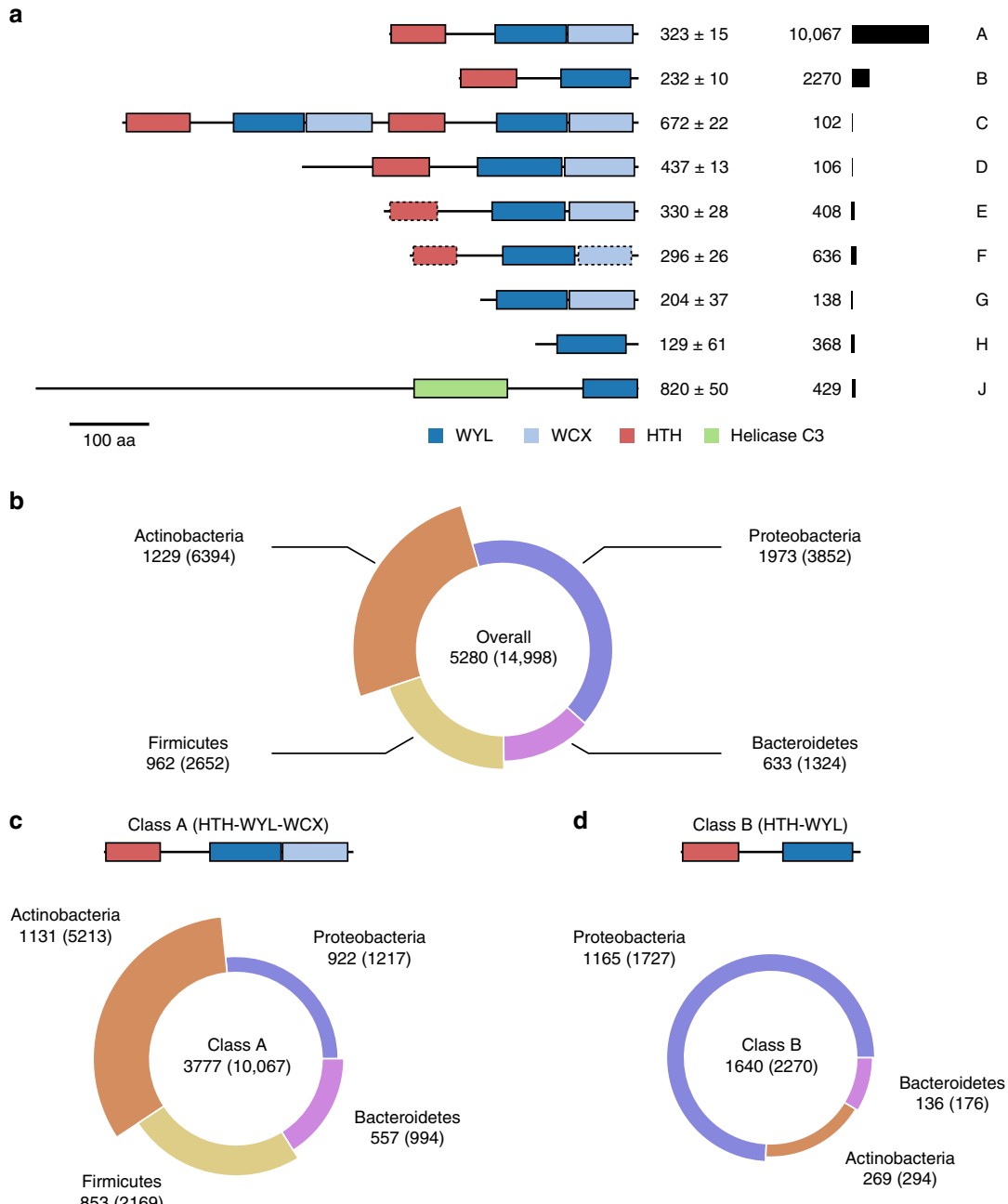

**Fig. 6** Domain architectures and taxonomic distribution of WYL domain-containing proteins. **a** Domain architecture classes of WYL domain-containing proteins reveal a tight association with an N-terminal HTH domain. Median and standard deviation of protein length are given next to the domain architecture sketch of each class followed by the number of sequences. Domain architecture sketches are drawn to scale based on the median values of protein length, domain boundaries, and domain length. The scale bar equals 100 amino acids (aa). Domains with dashed line borders were assigned manually. For clarity, architectures with less than 100 sequences are not shown. **b** The taxonomic distribution of all WYL domain proteins shows a prevalent occurrence in *Actinobacteria*. **c** Class A, featuring the WCX domain located C-terminally of the WYL domain, is mainly found in gram-positive bacteria, namely *Actinobacteria* and *Firmicutes*, while (**d**) Class B, exhibiting only the WYL domain, is mostly found in *Proteobacteria*. The segment radian represents the number of unique species, while the thickness of the segment represents the average number of sequences per species within that taxonomic group. The number of species is given below the class labels with the number of sequences in parentheses. For clarity, taxonomic groups smaller than 1.5% of the total number are not shown. See also Supplementary Fig. 11

## Discussion

During the mycobacterial DNA damage response, the heterodimeric transcriptional regulator PafBC activates most of the genes required for an adequate response to DNA stress[11,12]. However, understanding and experimentation concerning the regulatory mechanism of PafBC was largely hampered by a lack of knowledge about its molecular structure. This limitation has

manifested also in other studies concerning WYL domain-containing proteins[13–16]. Our computational analysis showed that roughly 90% of all WYL domain-containing proteins possess an N-terminal HTH domain suggesting that these are transcriptional regulators (Fig. 6a). Furthermore, our analysis revealed the WYL domain as a domain specific to bacteria that is present in nearly all bacterial phyla (Fig. 6b and Supplementary Fig. 11). Thus, it is

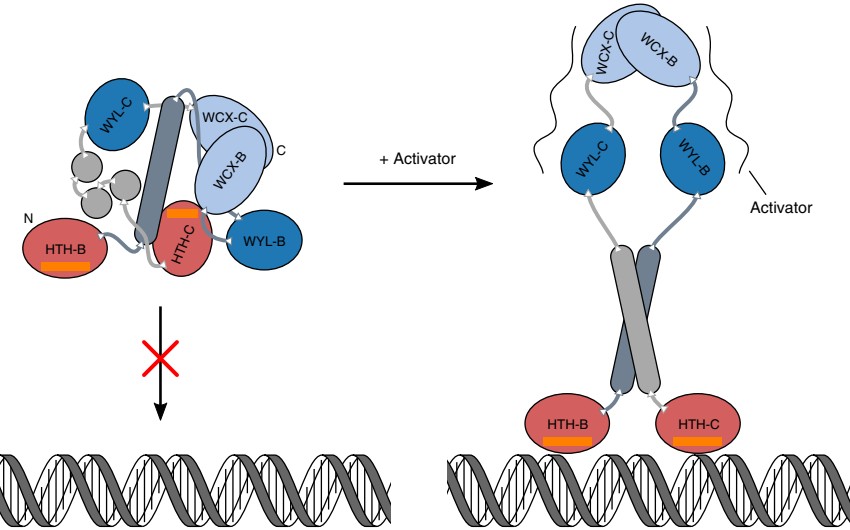

**Fig. 7** Hypothetical model of DNA binding by activated PafBC. In the non-activated state, PafBC buries the recognition helix (orange) of PafC's HTH domain (HTH-C) and cannot bind to its cognate promoter motif. Upon activator binding, PafBC likely undergoes large structural rearrangements of its domains to release the HTH-C domain, allowing promoter recognition and transcriptional activation of DNA repair genes

conceivable that the regulatory mechanism employed by PafBC might represent a shared mode of action for all of these regulators. This is not only an exciting possible concept, but might also be helpful in gaining a full understanding of the nature of the regulation.

We obtained the crystal structure of PafBC in the absence of DNA, in a largely asymmetric domain arrangement that is likely characteristic for the non-activated state. The domains are connected through long loops, suggesting a great degree of flexibility for the entire protein and that the protein might undergo large domain movements upon DNA binding, where it could eventually adopt a more symmetric arrangement, as observed for the WCX domains (Fig. 7). In such a state, the helices connecting HTH-C and WYL-C (α4', α4", α4"') could merge into a single helix and act as the counterpart to PafB helix α4 in a coiled-coiled fashion at the center of the protein. Such an interaction could be mediated by the row of hydrophobic residues that are featured along the axes of helices α4'-α4"'. Also, the HTH-C domain was observed in a state inaccessible for DNA binding (Fig. 2b). Besides their role in protein-DNA interaction, winged HTH domains were found to mediate protein-protein interactions[29–31]. Thus, the conformation of HTH-C may represent a state that is part of a regulatory mechanism, in which PafBC is prevented from efficient DNA binding under non-stress conditions. In agreement with this notion, the PafBC mutant lacking HTH-C cannot complement the phenotype of ΔpafBC observed under DNA stress (Fig. 5a), suggesting that HTH-C must fulfill an essential function, i.e., DNA binding/recognition. Furthermore, the recognition helices in the HTH domains of PafB and PafC are different in length and also in amino acid composition (Supplementary Fig. 4a and Supplementary Fig. 5), and likewise the PafBC binding motif (RecA-NDp) is non-palindromic[12]. Together with the regulatory switch, this could then also explain why PafBC is a heterodimer.

Our results provide key insights into the WYL and WCX domains, revealing that they adopt folds similar to proteins associated with RNA binding (Fig. 3, Fig. 4 and Supplementary Fig. 7). Considering that the activation of PafBC upon DNA damage does not rely on changes in protein levels[11], the possibility of another, stress-dependent factor required to elicit PafBC activity is likely. The results obtained in the complementation study with single amino acid substitutions in the WYL domains

strongly suggest a binding interface for a signal-transducing ligand (Fig. 5). Such a potential factor may thus well be an RNA molecule or, in a broader context, a nucleic-acid or nucleic acid derivative, relaying the signal of DNA damage to PafBC by recognition at the WYL and/or WCX domains. In fact, the WYL domain of PIF1 helicase from *Thermotoga elfii* was shown to bind single-stranded DNA, thereby stimulating helicase activity[15]. Binding of single-stranded DNA may be conceivable for the WYL domain proteins of class J of our computational analysis, which are also associated with a helicase domain. A potential RNA-based activating ligand for PafBC might originate from damaged RNA or from abortive transcripts at stalled RNA polymerase complexes that get released when transcription-repair coupling factor Mfd disrupts the transcription complex[32]. Furthermore, a vast amount of small RNAs was identified in mycobacteria, although none of them has been implicated in the DNA damage response yet[33–40]. In *Corynebacterium glutamicum*, the small 6C RNA, which is highly conserved in the actinobacterial phylum, was shown to increase about two-fold upon MMC treatment[41]. However, upregulated genes in response to 6C RNA overexpression in *M. tuberculosis* did not include DNA repair genes[42].

The Sm-fold of the WYL domain is characteristic of eukaryotic RNA-binding proteins, the Sm proteins. Their ring-shaped assemblies are core components of the spliceosomal snRNPs (small nuclear ribonucleoproteins)[43]. Through analogy, the bacterial protein Hfq is considered the sole representative of the Sm-like/LSm family based on its hexameric assembly state and RNA chaperone function[44]. Interestingly, no Hfq homolog has been identified in actinobacteria to date using sequence searches[45], but we found WYL domain-containing proteins to be significantly enriched in the actinobacterial phylum (Fig. 6b). It is possible that some of these actinobacterial WYL domain-containing proteins carry out a similar function to Hfq.

The crystal structure of PafBC provides the framework for understanding the mechanism by which PafBC connects the signal of DNA stress with a transcriptional response through use of its WYL/WCX domains. These results will also help us to better understand WYL domain-containing proteins in general.

## Methods
**Expression and purification of *Arthrobacter aurescens* PafBC.** Full-length <sup>Aau-</sup>PafBC was amplified from genomic DNA of *Arthrobacter aurescens* strain 579

(DSM-20116) using the primers aaubc-fw and aaubc-rv (Supplementary Table 1), which were designed based on the sequenced genome of strain TC1 (NC_008711; locus tag AAur_2182). The amplicon was cloned into a temporary vector and a truncated variant of $^{Aau}$PafBC ($^{Aau}$PafBCΔNC) missing the first 17 amino acids at the N-terminus and the last 7 amino acids at the C-terminus was amplified from this vector using primers aaubcdNC-fw and aaubcdNC-rv (Supplementary Table 1). The amplicon of $^{Aau}$PafBCΔNC was cloned into an isopropyl-β-D-thiogalactopyranosid (IPTG)-inducible expression vector with a cleavable His$_6$-TEV tag at the N-terminus. Selenomethionine-labeled protein was expressed according to a procedure adapted from ref. [46]: E. coli Rosetta (DE3) cells harboring the expression vector were grown as shaking cultures at 37 °C in M9 medium (M9 salts supplemented with 2 mM MgSO$_4$, 0.1 mM CaCl$_2$, 0.5% w/v glucose, 2 mg/l biotin, 2 mg/l thiamine, 0.03 mg/l FeSO$_4$). At an OD600 of 0.5, 100 mg/ml of phenylalanine, lysine, and threonine, 50 mg/ml of isoleucine, leucine, and valine, as well as 80 mg/ml of selenomethionine (Chemie Brunschwig) were added as solid powder to the cultures, which were further incubated for 30 min. Expression was then induced with 0.5 mM IPTG and cells were further incubated at 16 °C overnight. Cells were harvested (F9S, 7000 rpm (9180 × g), 10 min, 4 °C) and pellets were resuspended in lysis buffer (50 mM HEPES-NaOH pH 7.8/4 °C, 300 mM NaCl, 2 mM TCEP). The cell suspension was homogenized using a Heidolph DIAX600 mixer and cells were lysed by high pressure shear force using a Microfluidizer M110-L device (Microfluidics; 5 passes, 11,000 psi chamber pressure). After removal of cell debris (SS34, 20,000 rpm (47,810 × g), 4 °C, 30 min), the cleared lysate was supplemented with 1 mM PMSF, 1× c0mplete EDTA-free protease inhibitors (Roche), 50 U/ml DNase I, 10 mM imidazole and incubated for 30 min on ice. The lysate was passed over a self-packed Ni$^{2+}$-charged IMAC Sepharose 6 Fast Flow (GE Healthcare) column, and bound protein was eluted step-wise with lysis buffer containing 80 mM to 250 mM imidazole. After pooling protein-containing elution fractions, His-tagged TEV protease was added to a 1:30 molar ratio and the protein sample was dialyzed against 25 mM HEPES-NaOH pH 7.8/4 °C, 150 mM NaCl, 2 mM DTT, 1 mM EDTA at 4 °C overnight. TEV protease was removed by affinity chromatography and the protein sample was dialyzed against 25 mM HEPES-NaOH pH7.8/4 °C, 40 mM NaCl, 2 mM DTT, 1 mM EDTA at 4 °C overnight. The protein was further loaded on a Source 30Q column and eluted with a 50 mM to 400 mM NaCl gradient in 25 mM HEPES-NaOH pH 7.8/4 °C, 1 mM TCEP. Protein-containing elution fractions were pooled and concentrated using an Amicon Ultra 30 K centrifugal filter (3500 × g, 4 °C). The concentrated protein was run on a self-packed 100 ml Superose 12 prep grade column in crystallization buffer (10 mM HEPES-NaOH pH 7.8/4 °C, 50 mM NaCl, 1 mM TCEP, 0.1 mM EDTA). Protein was concentrated as above to 22 mg/ml, aliquotted, frozen in liquid nitrogen and stored at −80 °C until use.

**Crystallization of Arthrobacter aurescens PafBC**. Crystals of $^{Aau}$PafBCΔNC were grown in sitting drop vapor diffusion plates (Hampton Research) at 4 °C by mixing 1–2 μl protein solution (5.6–7.5 mg/ml) and 1 μl reservoir solution. Crystals appeared after 2–3 days using reservoir solutions containing 100 mM Bis-Tris-propane pH 8.8 to 9.2 (20 °C), 80–160 mM KSCN, 18–21% PEG-2000 (Sigma). PEG-2000 was added in 5% steps to reach a final concentration of 36% w/v in drops containing crystals before flash cooling the crystals in liquid nitrogen.

**Data collection, structure determination, and refinement**. Reflection image data was collected at the X06SA beamline of the Swiss Light Source (SLS, Paul-Scherrer-Institut, Villigen, Switzerland) at 100 K and 12,670 eV beam energy (0.978561 Å). Diffraction images were processed using XDS[47] and scaled using AIMLESS[48]. Determination of heavy atom sites, initial phases, and crude main chain tracing were carried out using the SHELX programs[49]. The resulting experimental electron density map displayed easily discernible protein features. The initial model from SHELX was further extended with PHENIX AutoBuild[50] and subsequent iterative model building and refinement was carried out using Coot[51] and phenix.refine[52], respectively.

**Structure visualization**. Graphical representations of protein structures were prepared using UCSF Chimera v1.12 build 41623[53]. Amino acid residue conservation was calculated using AL2CO (independent counts, BLOSUM-62 matrix)[54]. For that, PafBC protein sequences of 23 different organisms spanning the entire actinobacterial phylum (UniProt accession numbers P9WIM1, I7G3U5, C0ZZU3, A1SK18, A7BCC5, A4X749, A0LU62, A6W976, Q8NQE2, Q9RJ64, H6RJ02, D2ATU2, C8XAP4, C7PVW0, A0A160VN40, Q0RLT0, A0A1D7W444, A0A1H2KTF9, B2GIN6, Q47P13, C5CBV3, A9WSH6, A1R6R2, P9WIL9, A0QZ41, C0ZZU2, A1SK19, A7BCC4, A4X750, A0LU63, A6W977, Q8NQE3, Q9RJ65, H6RJ01, D2ATU1, C8XAP3, C7PVW1, A0A161KHT8, Q0RLS9, A0A1D7W495, A0A1H2KTV3) were aligned against the A. aurescens strain 579 PafBC protein sequence. To be able to calculate conservation across the entire length of naturally fused PafBC proteins, sequences of separately encoded PafB/PafC proteins were concatenated prior to alignment.

**Mutational screening of Mycobacterium smegmatis PafBC**. The coding sequence of pafBC was amplified from Mycobacterium smegmatis mc2-155 SMR5[55] genomic DNA and cloned into a pMyNT-derived integrative plasmid (pMyNT template provided by A. Geerlof, EMBL Hamburg; for primer sequences, see Supplementary Table 1). As promoter, a DNA fragment containing 347 bp upstream of the pafA coding sequence was inserted in front of pafBC to generate the pafBC complementation plasmid. Variants were then generated by KLD site-directed mutagenesis or by Gibson assembly using the pafBC complementation plasmid as template. A control vector was created by amplification and re-ligation of the pMyNT-derived integrative plasmid using primers pMyNTint-fw and pMyNTint-rv (Supplementary Table 1). The various plasmids were then transformed into the Mycobacterium smegmatis ΔpafBC strain[11] (the empty control vector was transformed into the wild-type and the ΔpafBC strain), and viability in presence of mitomycin C was assessed with the resazurin assay as described previously[12]. In addition, viability was assessed by a spotting assay: Cells were grown as shaking cultures in 7H9 medium supplemented with 50 μg/ml hygromycin at 37 °C. At an OD600 of 1.0 to 1.3, cultures were split and mitomycin C was added to one half to a final concentration of 80 ng/ml. Cultures were incubated for another 4 h at 37 °C. Subsequently, cultures were diluted to an OD600 of 0.8 based on the OD600 value measured at the time of splitting and a 1:5 serial dilution was prepared in 7H9 supplemented with 50 μg/ml hygromycin. 10 μl were spotted onto 7H10 agar plates and the plates were incubated facing upwards at 37 °C for 3 days.

**Immunoblotting**. M. smegmatis strains carrying complementation plasmids or empty control vectors were grown as shaking cultures to an OD600 of about 1.0 in 7H9 medium supplemented with 50 μg/ml hygromycin at 37 °C. Cells were harvested by centrifugation (3000×g, 5 min) and were washed once in 700 μl PBS. Pellets were then resuspended in 700 μl PBS supplemented with 1 mM DTT, 1 mM EDTA, 1 mM PMSF, 1× cOmplete EDTA-free protease inhibitor (Roche) and lysed by bead beating in 2 ml screw cap tubes containing 500–700 mg 0.15 mm zirconium oxide beads using an MP Biomedicals FastPrep-24 bead beater (2 × 30 s, 6 m/s, 1 min pause on ice in between). After removing insoluble material by centrifugation (16,000 × g, 5 min, 4 °C), protein content of the cleared lysate was determined by Bradford assay. Equal amounts of total protein were separated on a 10% SDS-containing polyacrylamide gel and transferred onto PVDF membrane Immobilon-P (Merck) using a BioRad Trans-Blot SD semidry transfer cell (20 V, maximum of 10 mA/cm$^2$, 30 min). Membranes were blocked in PBS containing 0.05% Tween-20 and 1% polyvinylpyrrolidone 40 (PVP-40; Sigma) for 1 h at room temperature. Immunoblotting for RecA, RpoB, or PafBC was carried out using commercially available anti-RecA antibody (1:2000, MBL International, clone ARM414) raised in mouse, commercially available anti-RpoB antibody (1:5000, BioLegend, clone 8RB13) raised in mouse, or previously described anti-PafBC antibody (1:5000) raised in rabbit[11]. The blocked membrane was incubated with primary antibodies in blocking buffer for 1 h at room temperature. Detection was achieved by incubation with horseradish peroxidase conjugated anti-mouse IgG antibody (1:250,000, abcam #ab6789) or anti-rabbit IgG antibody (1:10,000, Invitrogen #65-6120) in blocking buffer for 1 h at room temperature and using ECL substrate (BioRad Clarity Western Substrate).

**Computational analysis of WYL domain-containing domains**. All alignments were generated using ClustalO v1.2.4 with default settings[56]. Manual analyses of alignments were performed in Jalview v2.10.5[57]. Steps involving hidden Markov-models (HMMs) were conducted using the programs "hmmbuild", "hmmsearch", and "hmmscan" from the software suite HMMER v3.2.1 (hmmer.org)[28]. A local search database was created from all UniProt reference proteomes (September 2018 release, uniprot.org)[58].

For the initial analysis, PafBC homologs from other actinobacteria were identified by BLAST (blast.ncbi.nlm.nih.gov; restricted search to actinobacterial species, otherwise default settings)[59] using Mycobacterium smegmatis PafB or PafC as input. The identity of the obtained PafBC homologs was cross-checked on the genome level for the operon organization of the genes and association with the Pup-proteasome system gene locus. In total, ten PafB/PafC sequence pairs were retrieved (UniProt accession numbers P9WIM1, P9WIL9, I7G3U5, A0QZ41, A7BCC5, A7BCC6, A4X749, A4X750, C0ZZU3, C0ZZU2, A1SK18, A1SK19, A0LU62, A0LU63, A6W976, A6W977, Q8NQE2, Q8NQE3, Q9RJ64, Q9RJ65). A global alignment of these sequences was used to generate an HMM with "hmmbuild" (default settings), which was subsequently used to search against the reference proteomes database with "hmmsearch" (command line options were -E 1 -domE 1 -incE 0.01 -incdomE 0.03). A list of domains contained in the retrieved sequences was obtained with the "hmmscan" module (command line options were -E 0.1 -domE 0.1 -incE 0.01 -incdomE 0.03) using all HMMs from the Pfam database release 32.0[60]. Domain lengths were analyzed based on the envelope boundaries given by "hmmscan" for sequences with a domain score above 30.0 (independent domain e-value < 0.001).

Because the Pfam HMM profile of the WYL domain (Pfam-WYL) includes both WYL and WCX domain, we had to define a new WYL domain HMM in order to annotate it correctly for our analysis. To build the WYL HMM, 250 sequences were randomly sampled from sequences with a Pfam-WYL length greater than 127 residues and another 250 sequences were randomly sampled from sequences with a Pfam-WYL length less than 127 residues (other thresholds: domain score > 30.0, independent domain e-value < 0.001). The threshold was chosen based on the domain boundaries seen in the crystal structure of $^{Aau}$PafBC and the length

distribution of C-terminal region of the retrieved Pfam-WYL-containing proteins. Sequences with obvious defects in the WYL region were discarded manually resulting in 477 entries that were used for alignment. From this alignment, the WYL domain boundaries were established using the N-terminal boundary of the Pfam-WYL, while the C-terminal boundary was chosen by the C-termini of short Pfam-WYLs and the crystal structure. The alignment was then trimmed to the WYL boundaries, sequences with a pairwise identity above 70% were clustered using CD-HIT v4.6.8 (command line options -n 4 -c 0.7)[61] and an HMM was generated as above. To mature the WYL HMM, an iterative approach similar to the Pfam HMM generation was chosen. Sequences were retrieved from the reference proteomes using the WYL HMM to generate a full alignment, which was then again trimmed to the WYL domain boundaries to make up a new seed alignment used to build a new HMM (gathering threshold: domain score > 27.0). The process was repeated for a total of three iterations.

To build the HMM for the unrecognized (winged) HTH domains in PafBC and many other WYL domain-containing proteins, 500 sequences were randomly sampled from the group of sequences with a median length of 326 amino acids and containing only a WYL domain (thresholds: domain score > 30.0, independent domain e-value < 0.001). The sequences were aligned and curated based on the presence of the conserved blocks representing the helices of the HTH fold. The sequences were further split into groups exhibiting a PafB-like or PafC-like wing sequence of 184 and 235 sequences, respectively. For each group, an HMM was built and searched against reference proteomes database as described above, but using an identity threshold of 90% and iterating only once in order to keep a separation to other HTH-type domains.

To build the HMM for the C-terminal extension found in many WYL domains (WCX), 500 sequences were randomly sampled from the group containing a Pfam-WYL with a length greater than 127 amino acids (thresholds: domain score > 30.0, independent domain e-value < 0.001). Sequences missing the C-terminal conserved block were removed, leaving 434 sequences for HMM generation. The HMM generated from the manually curated seed alignment was used without iteration.

The custom HMMs were then added to the local search database of the Pfam HMM database (see also above).

To establish the domain architecture classes of the WYL domain-containing proteins, the WYL HMM was used to search against the reference proteome database with "hmmsearch" (command line options as above). Obtained sequences with a domain score > 27.0 were analyzed for the presence of other domains using "hmmscan" together with the local Pfam HMM database including the custom HMMs as described above. Protein sequences were then categorized according to their sequence length, their identified domains (on the level of Pfam clans; independent domain e-value < 0.01) and the domain length. Domain architecture classes with large regions apparently containing no domain were checked manually by alignment for presence of conserved features, by alignment to other classes and by alignment to the manually curated PafBC seed sequences (see above). They were then grouped together with other classes, where appropriate. Due to low overall abundance and being mostly candidate species, all non-bacterial sequences were excluded from the main analysis.

**Reporting summary**. Further information on research design is available in the Nature Research Reporting Summary linked to this article.

## Data availability
Protein structure data and coordinates of this study have been deposited in the Protein Data Bank (PDB) with the accession code 6SJ9. Source data for Fig. 5, Supplementary Fig. 8, and Supplementary Fig. 9 are provided with the paper. Other data are available from the corresponding author upon reasonable request.

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

## Acknowledgements

We thank Takashi Tomizaki and the staff of beamline X06SA at the Swiss Light Source (Villigen, Switzerland) for support with data collection; Beat Blattmann and Céline Stutz-Ducommun of the Protein Crystallization Center (University of Zurich) for support with the initial screens; Marcel Bolten for support with X-ray data analysis. The research was funded by the Swiss National Science Foundation (SNSF), grant no. 31003A-163314.

## Author contributions

A.U.M., N.B., and E.W.B. designed research and analyzed data. A.U.M. performed protein purification, crystallization, and in vivo experiments. A.U.M. and M.L. analyzed crystallographic data. A.U.M., N.B., and E.W.B. wrote the paper. All authors contributed to editing of the paper.

## Competing interests

The authors declare no competing interests.
