## [Peer Review File · Nature Communications]

Reviewers' comments:

Reviewer #1 (Remarks to the Author):

This manuscript is an important contribution that examines the structure of the PafBC heterodimeric transcription factor. Important recent data from this same group indicates that PafBC mediates the so called "recA independent", ie the non-SOS arm, of the DNA damage response in mycobacteria. In contrast to SOS, which is repressed by the LexA repressor until RecA catalyzed cleavage relieves this repression, the PafBC complex appears to be an activator based on genetic studies. The proteins are present without DNA damage and therefore the activation mechanism for PafBC in the DDR is unknown, but a very important question.

In this manuscript, the authors solve the structure of a natural PafBC fusion protein from *Arthrobacter* and use the structural features to deduce structural information about actinobacterial homologues. As the authors show, this class of proteins is widely distributed and therefore the conclusions and potentially significant. The major conclusions are:

- 1) PafBC is an asymmetric dimer in which the HTH DNA binding domain of PafC is somewhat buried is what is presumed to be the unactivated state. This suggests that an activation event results in reorientation of the HTH to bind DNA, although this state is not captured as attempts to reconstitute in vitro DNA binding to the promoter element have not been successful.
- 2) the C terminus of PafB and PafC contain a domain that by structural homology is similar to RNA binding proteins such as Hfq.
- 3) functional complementation of an *M. smegmatis* pafBC null strain with alanine substitution mutations at conserved residues implicates some of these C terminal residues in PafBC function.

Comments and suggestions:

1) Given the importance of PafBC in the mycobacterial DDR, and the broad distribution of this protein family, the structure of the *Arthrobacter* protein provides new insight into the protein family and implies that it may bind RNA. The implication of this structural finding is that the inducing signal for the PafBC pathway may be an RNA ligand, which is potentially very significant and novel. However, the study does not actually demonstrate RNA binding in vitro, nor does it provide data that this RNA is actually the activating signal in vivo, despite the functional importance of some of the residues predicted to be involved in RNA binding. I realize this is a major next step in the work, but the functional extension of the important new structural data is more of a hypothesis rather than a finding, somewhat limiting the broad impact. Some speculation in the discussion about how an RNA ligand would be generated to activate PafBC during the DDR would be helpful. Do the authors hypothesize that the RNA would originate from damage in actively transcribed genes? From RNA generated from another source?

2) Some clarification is required for the in vivo complementation experiments. In the genome sequence of *M. smegmatis* mc2155, a frameshift mutation is reported in pafB (see <https://www.ncbi.nlm.nih.gov/gene/4534207>) . By my analysis of this sequence, this frameshift appears to truncate PafB to a size of approximately 27kD due to a premature stop codon, but with the addition of a neo C terminus between the frameshift and the stop. The preserved native sequence of PafB in the frameshifted ORF ends at the glycine of the VEPWG (amino acid 208 in the alignment in figure S2). The predicted size of the intact, non frameshifted PafB is 36kD, slightly larger than PafC at 34kD. This would suggest that the sequenced strain of mc2155 could be have a partially functional PafB or PafBC, given that this frameshift removes the WYL and WCX domains. It is therefore critical to determine whether the mc2155 derivative used in this study contains this mutation, and whether the complementation constructs used in the experiments also contain this mutation. The western blots in Figure 5, using an antibody to both PafB and PafC, detect two proteins of different sizes, and the HTH deletion of PafC in panel G confirms that the smaller of these two bands is PafB, suggesting it may be truncated. Although seemingly a detail, the authors should clarify this point as it would have broad implication for interpretation of the complementation data.

3) Assuming the issue in point #2 is not present, then the experiments in figure 5 show an additive effect of PafB and PafC mutations on mitomycin sensitivity. However, in some cases (e.g. B F210A/CF200A) the protein is unstable and therefore the damage sensitivity is not specific to a functional effect of that mutation.

4) The data in figure 5 is derived from resazurin assays and the paper refers to this as "viability". Resazurin is a redox indicator that reflects cellular metabolism and is not indicative of viability. A more traditional method of assessing cellular viability would be preferable, such as dilution spotting for survival. I think it would also strengthen the paper to assess the functionality of the mutations on the efficiency of DNA damage induced, PafBC dependent, transcription. The prediction is that mutations in the C terminal domains would affect the efficiency of transcriptional induction according to the model. Cellular viability is only an indirect measure of this function, especially since PafBC has a role in maintaining RecA levels in the basal state, as these authors have nicely demonstrated previously.

Minor points:

1) I don't think it is useful for the field to try to establish a war of primacy of PafBC vs SOS by asserting that PafBC is quantitatively more important (ie statements of more genes being controlled). One could just as easily assert that the literature says that ALL mutagenesis, the ultimate cause of antibiotic resistance, is SOS dependent. Both systems are important and interesting.

Reviewer #2 (Remarks to the Author):

Two pathways have been described by which bacteria activate their DNA repair machinery in response to genotoxic stress. The first is the SOS response which leads to the removal of the LexA repressor and thereby to the expression of the SOS genes. In the second pathway the either heterodimeric or in some species fused PafBC protein acts as a transcriptional activator upon genotoxic stress. However, the mode of activation is so far not clear. The authors therefore pursued structural studies on the PafBC protein from *Arthrobacter aurescens* and used this structure as a basis to delineate its mode of action. The structure was solved in its apo form that seems to represent an auto-inhibited state since one of the DNA binding domains is involved in protein interactions and would not be accessible for DNA interactions. The authors show that the structure can be clearly divided into three distinct domains that are present in both PafB and PafC: the winged helix-turn-helix domain (HTH), the WYL domain and the C-terminal extension of the WYL domain (WCX). The latter two domains were previously just described as the WYL domain but importantly the C-terminal WCX domain seems to be essential for dimer formation whereas the WYL domain assumes a similar fold as the Hfq protein. Due to the similarity to the Hfq protein the authors speculate that RNA may be required for the activation of the PafBC protein. The authors then proceed with a mutational complementation screen to delineate the importance of specific domains and amino acids and finally analyze the distribution of WYL domain containing proteins. Overall this is a very interesting analysis and should be published in Nature communications after the following points have been addressed:

1. Since the authors suggest that RNA could be required for the activation of the PafBC protein the question immediately arises whether any attempts have been pursued to test this hypothesis?

Minor points:

1. The title suggests that the structure of the mycobacterial protein was solved, which is not the case. Maybe the authors could adjust the title so that this is not implied.
2. Important residues such as F42 and F46 or the two arginines which have also been mutated are just shown in Figure S4B. Please move this Figure in to the main Figures since these residues seem to play a major role in PafBC.
3. On page 7 the authors write that deletion of either HTH domain leads to the same reduced viability as observed for the delta pafBC strain. Even though the authors then continue with the

statement that the delta HTH-B variant could not be readily expressed they should not initially state that they can make this conclusion. They can only draw a conclusion on the delta HTH-C variant.

4. On page 9 the authors claim that roughly 90% of all WYL domain containing proteins possess an N-terminal HTH domain which suggests that these proteins are all transcriptional regulators. Is this assumption going too far? Can't they just be DNA binding proteins without being a transcriptional regulator?

5. In Figure 4C numbers are indicated above the protein sequence (in other Figures as well). Which numbers do they represent? The numbering of the different proteins in the sequence alignment is most likely not the same for all the proteins?

6. In Figure 4C it would be nice if the secondary structure elements are numbered.

Reviewer #3 (Remarks to the Author):

Muller et al, Structure and functional implications of WYL domain-containing transcription factor PafBC involved in the mycobacterial DNA damage response.

The PafBC transcription factor is responsible for upregulating the majority of DNA damage response genes in mycobacteria. The authors have determined the crystal structure of the PafBC protein from *Arthrobacter aurescens*. They have identified two modules in the protein each having an N-terminal HTH motif and C-terminal WYL and WCX domains. The structure is rather exciting in that it shows an asymmetric domain arrangement in the non-activated state. The authors posit that binding of an activating ligand (possibly ss nucleic acid) would induce a structural rearrangement to allow the two HTH domains to bind the promoter. They also performed mutational complementation assays to show residues in the WYL domain are essential for function. Finally, they carried out an extensive bioinformatics analysis of WYL domain-containing proteins to determine their functional context and distribution.

Overall this is a well written and interesting manuscript. From the statistics provided, the structure appears to be well determined. The figures are clearly presented and the conclusions are logical. This work will have broad appeal to investigators interested in DNA damage and repair, microbiology, or transcriptional regulation.

Point-by-point answers:

We thank the reviewers for their time and constructive comments. These are our point-by-point answers.

Reviewer #1:

1) Given the importance of PafBC in the mycobacterial DDR, and the broad distribution of this protein family, the structure of the Arthrobacter protein provides new insight into the protein family and implies that it may bind RNA. The implication of this structural finding is that the inducing signal for the PafBC pathway may be an RNA ligand, which is potentially very significant and novel. However, the study does not actually demonstrate RNA binding in vitro, nor does it provide data that this RNA is actually the activating signal in vivo, despite the functional importance of some of the residues predicted to be involved in RNA binding. I realize this is a major next step in the work, but the functional extension of the important new structural data is more of a hypothesis rather than a finding, somewhat limiting the broad impact. Some speculation in the discussion about how an RNA ligand would be generated to activate PafBC during the DDR would be helpful. Do the authors hypothesize that the RNA would originate from damage in actively transcribed genes? From RNA generated from another source?

As the reviewer remarks, the identification of the response producing ligand is the next major step. Our structural analysis provides some interesting leads that we now need to follow up on. We have added text in the discussion section, mentioning possible RNA ligand sources. However, since, as the reviewer correctly points out, the identity of the ligand remains unknown, we do not want to extend this discussion too much.

2) Some clarification is required for the in vivo complementation experiments. In the genome sequence of *M. smegmatis* mc2155, a frameshift mutation is reported in *pafB* (see <https://www.ncbi.nlm.nih.gov/gene/4534207>). By my analysis of this sequence, this frameshift appears to truncate PafB to a size of approximately 27kD due to a premature stop codon, but with the addition of a neo C terminus between the frameshift and the stop. The preserved native sequence of PafB in the frameshifted ORF ends at the glycine of the VEPWG (amino acid 208 in the alignment in figure S2). The predicted size of the intact, non frameshifted PafB is 36kD, slightly larger than PafC at 34kD. This would suggest that the sequenced strain of mc2155 could have a partially functional PafB or PafBC, given that this frameshift removes the WYL and WCX domains. It is therefore critical to determine whether the mc2155 derivative used in this study contains this mutation, and whether the complementation constructs used in the experiments also contain this mutation. The western blots in Figure 5, using an antibody to both PafB and PafC, detect two proteins of different sizes, and the HTH deletion of PafC in panel G confirms that the smaller of these two bands is PafB, suggesting it may be truncated. Although seemingly a detail, the authors should clarify this point as it would have broad implication for interpretation of the complementation data.

*The reviewer is indeed correct that the reference genome cited above reports a frameshift for the *pafB* gene. However, the reported frameshift is an error present in this specific reference genome for the following reasons:*

a) Other genome assemblies for M. smegmatis MC2 155 do not exhibit this frameshift. See NCBI accession numbers NC_018289.1 (locus tag MSMEI_3799, protein sequence WP_003895340.1) and NZ_CP009494.1 (locus tag LJ00_19320, protein sequence WP_003895340.1).

b) We have amplified the region in question from the wild-type M. smegmatis strain used in the complementation experiments (MC2 155 SMR5) and sequenced the PCR product. No frameshift was detected.

c) We have amplified the pafB gene from the M. smegmatis MC2 155 laboratory strain and have cloned it into a protein expression plasmid. The construct was subsequently sequenced and also did not exhibit the frameshift.

d) Our previous publication on PafBC (Müller et al (2018) Cell Reports 23:12, 3551-3564) included a ChIP-seq experiment, where the genome of the M. smegmatis mc2-155 strain was sequenced as background control. Assembly of the consensus sequence from the next-generation sequencing reads confirms the presence of a full-length pafB gene (i.e. no frameshift). Data are available on ArrayExpress, accession number E-MTAB-6503.

e) The pafB gene product of M. smegmatis has already been described in two other publications, which also did not report any frameshift or truncations: Olivencia et al (2017) Scientific Reports 7: 13987 and Korman et al (2018) Fut. Microbiol. 14, 11-21.

3) Assuming the issue in point #2 is not present, then the experiments in figure 5 show an additive effect of PafB and PafC mutations on mitomycin sensitivity. However, in some cases (e.g. B F210A/CF200A) the protein is unstable and therefore the damage sensitivity is not specific to a functional effect of that mutation.

Amongst the 15 pafBC single or double mutants used in the complementation analysis, only one showed bands in the α PafB or α PafC Western blot lower than what is observed for the wild type strain, namely the double mutant mentioned above (B-F210A/C-F200A). We decided to nevertheless include this double-mutant. For one, the low band intensity could be due to an effect of the mutation on the recognized epitopes. Furthermore, by showing the blot, we lay open the data instead of simply leaving out that mutant. However, to alert the reader to this potential problem for one of the 15 mutant complementations, we are now explicitly mentioning the low band intensity and potentially low expression level for this mutant in the results section text.

4) The data in figure 5 is derived from resazurin assays and the paper refers to this as “viability”. Resazurin is a redox indicator that reflects cellular metabolism and is not indicative of viability. A more traditional method of assessing cellular viability would be preferable, such as dilution spotting for survival. I think it would also strengthen the paper to assess the functionality of the mutations on the efficiency of DNA damage induced, PafBC dependent, transcription. The prediction is that mutations in the C terminal domains would affect the efficiency of transcriptional induction according to the model. Cellular viability is only an indirect measure of this function, especially since PafBC has a role in maintaining RecA levels in the basal state, as these authors have nicely demonstrated previously.

As suggested by the reviewer, we have carried out dilution spotting survival assays for the complemented strains presented in Figure 5 a-f. The dilution spotting tests agree well with the reported resazurin indicator curves and have been added as a separate figure to the Supplement (new Supplementary Figure S10).

We have furthermore assessed the level of RecA expression obtained in response to complementation with the various PafBC mutant proteins. The mutations in the C-terminal domains result in defective upregulation of RecA protein levels in the cell. These data have been included in Figure 5.

Minor points:

1) I don't think it is useful for the field to try to establish a war of primacy of PafBC vs SOS by asserting that PafBC is quantitatively more important (ie statements of more genes being controlled). One could just as easily assert that the literature says that ALL mutagenesis, the ultimate cause of antibiotic resistance, is SOS dependent. Both systems are important and interesting.

It is certainly not our intention to establish any ranking of importance. However, the SOS response is the main established DNA response pathway in the literature despite the fact that in mycobacteria a large majority of genes are upregulated independent of the SOS response and are instead regulated by PafBC. We feel it is important to stress this fact precisely because the SOS response is so predominant in the literature. Both pathways are moreover connected by the fact that RecA, a main regulatory player in the SOS response, is upregulated by PafBC. Clearly the two response pathways are intertwined and complementary. We have added a statement to this effect to the introduction.

Reviewer #2:

Overall this is a very interesting analysis and should be published in Nature communications after the following points have been addressed:

1. Since the authors suggest that RNA could be required for the activation of the PafBC protein the question immediately arises whether any attempts have been pursued to test this hypothesis?

Naturally, we have pursued attempts to identify the activating ligand of PafBC, even before we obtained the crystal structure. These were mainly targeted at rationally selected potential ligands, which included a number of RNA molecules, and were tested using electro-mobility shift assays (EMSA). Employing PafBC in typical binding assays (e.g. ITC) is complicated by the fact that PafBC contains two nucleic acid binding HTH domains. Our attempts with rationally selected ligands did not yield a positive result yet, which could have a multitude of reasons. Clearly, testing of our hypothesis and identifying the activating ligand are the next major steps we will pursue, in order to allow us to study PafBC's activation mechanism in detail. This will require a comprehensive, unbiased ligand search and is a large undertaking in itself. We have initiated this, but it will take time and effort well beyond the scope of this revision, unfortunately.

Minor points:

1. The title suggests that the structure of the mycobacterial protein was solved, which is not the case. Maybe the authors could adjust the title so that this is not implied.

It has only been shown for mycobacteria and not Arthrobacter that PafBC is involved in the DNA damage response, and this is reflected in the title. The manuscript includes both structural data on a homolog from Arthrobacter and functional analysis of the M. smegmatis system. As we are very restricted in the length of the title, we cannot go into such detail. However, the organism from which the PafBC structure was determined is mentioned prominently in the abstract. It was a challenge to come up with a title that fits the word count and that provides the bottom line of the study. We think our current title manages this balance very well and we would therefore like to keep it.

2. Important residues such as F42 and F46 or the two arginines which have also been mutated are just shown in Figure S4B. Please move this Figure in to the main Figures since these residues seem to play a major role in PafBC.

The mutated residues are shown in Figure 4e, which is a panel in the main part of the manuscript, not the supplement. The residues shown in the supplementary figure S4B are located in the helix-turn-helix motif of PafC and not the putative ligand-binding domain. As we have reached the maximum allowed number of figures for the document, these zoomed in structural views are shown in the supplement.

3. On page 7 the authors write that deletion of either HTH domain leads to the same reduced viability as observed for the delta pafBC strain. Even though the authors then continue with the statement that the delta HTH-B variant could not be readily expressed they should not initially state that they can make this conclusion. They can only draw a conclusion on the delta HTH-C variant.

We have reworded this section in the text according to the reviewer's recommendation. Additionally, we now provide data for an HTH-B mutant (new Supplementary Figure S9) where instead of domain deletion, two highly conserved arginines in the recognition helix were changed to alanines. This PafBC variant expresses well, but exhibits the characteristics of the pafBC knockout strain concerning viability and RecA induction, supporting the requirement for a functional HTH-B domain.

4. On page 9 the authors claim that roughly 90% of all WYL domain containing proteins possess an N-terminal HTH domain which suggests that these proteins are all transcriptional regulators. Is this assumption going too far? Can't they just be DNA binding proteins without being a transcriptional regulator?

The only investigated member, mycobacterial PafBC, changes the transcript level of 152 genes. Based on the preserved subunit arrangement and the sequence homology, the most likely scenario is that these proteins are transcriptional regulators. In absence of any data to the contrary, we feel that the most likely explanation should be put forth. It is clear from the text what we base our hypothesis on and it is worded carefully enough in our opinion.

5. In Figure 4C numbers are indicated above the protein sequence (in other Figures as well). Which numbers do they represent? The numbering of the different proteins in the sequence alignment is most likely not the same for all the proteins?

The numbering in the alignment reflects the alignment position numbering and not positions in any single one of the individual sequences. We have added this explanation to the figure legend.

6. In Figure 4C it would be nice if the secondary structure elements are numbered.

As suggested by the reviewer, we have now numbered the secondary structure elements for better orientation of the reader.

REVIEWERS' COMMENTS:

Reviewer #1 (Remarks to the Author):

The authors have responded convincingly to all of my prior comments.